# Structure of a type IV secretion system core complex encoded by multi-drug resistance F plasmids

Xiangan Liu[1,3], Pratick Khara[1,3], Matthew L. Baker[2], Peter J. Christie ◌ [1✉] & Bo Hu ◌ [1✉]

Bacterial type IV secretion systems (T4SSs) are largely responsible for the proliferation of multi-drug resistance. We solved the structure of the outer-membrane core complex (OMCC$_F$) of a T4SS encoded by a conjugative F plasmid at <3.0 Å resolution by cryoelectron microscopy. The OMCC$_F$ consists of a 13-fold symmetrical outer ring complex (ORC) built from 26 copies of TraK and TraV C-terminal domains, and a 17-fold symmetrical central cone (CC) composed of 17 copies of TraB β-barrels. Domains of TraV and TraB also bind the CC and ORC substructures, establishing that these proteins undergo an intraprotein symmetry alteration to accommodate the C13:C17 symmetry mismatch. We present evidence that other pED208-encoded factors stabilize the C13:C17 architecture and define the importance of TraK, TraV and TraB domains to T4SS$_F$ function. This work identifies OMCC$_F$ structural motifs of proposed importance for structural transitions associated with F plasmid dissemination and F pilus biogenesis.

[1] Department of Microbiology and Molecular Genetics, McGovern Medical School, 6431 Fannin St, Houston, TX 77030, USA. [2] Department of Biochemistry and Molecular Biology, McGovern Medical School, 6431 Fannin St, Houston, TX 77030, USA. [3]These authors contributed equally: Xiangan Liu, Pratick Khara. ✉email: Peter.J.Christie@uth.tmc.edu; Bo.Hu@uth.tmc.edu

The type IV secretion systems (T4SSs) are a diverse super-family of translocation systems found in many species of bacteria[1,2]. Functionally, T4SSs are classified mainly as conjugation machines or effector translocators[3]. The conjugation systems mediate the transfer of mobile genetic elements (MGEs) and their cargoes of antibiotic resistance genes and virulence determinants among bacteria[4]. The effector translocators instead deliver effector proteins and other macromolecules to eukaryotic cells to aid in the establishment of bacterial pathogenic or symbiotic interactions[5]. All T4SSs are assembled from a minimum set of signature subunits, which are homologs or orthologs of the 11 VirB and VirD4 components that build the prototypical T4SS used by *Agrobacterium tumefaciens* to deliver oncogenic T-DNA to plants[6]. T4SSs designated as minimized are composed only of VirB/VirD4-like subunits. Other T4SSs classified as expanded have acquired novel domains among the VirB-like subunits or appropriated as many as two dozen system-specific components of distinct ancestries presumably for specialized functions. The best-characterized of the expanded systems are the Dot/Icm effector translocator functioning in *Legionella pneumophila*, the Cag effector system in *Helicobacter pylori*, and the Tra conjugation machine encoded by IncF plasmids[2].

The T4SSs are organized as cell-envelope-spanning nanomachines with distinct architectural features that include an outer-membrane core complex (OMCC), an inner membrane complex (IMC), a connecting periplasmic stalk or cylinder, and two or three hexameric ATPases positioned at the cytoplasmic face of the IMC[1,7]. Conjugation machines and some effector translocator systems also elaborate extracellular pili that mediate attachment to abiotic and biotic surfaces and contribute to the formation of robust biofilms[8–10]. Although the IMCs are refractory to purification for structural analyses, OMCCs associated with two minimized (*Xanthomonas citri* VirB/VirD4, pKM101-encoded Tra) and two expanded (*L. pneumophila* Dot/Icm, *H. pylori* Cag) systems have been solved at or near-atomic resolutions[11–16]. OMCCs of minimized systems are composed only of homologs or orthologs of the *A. tumefaciens* VirB7, VirB9, and VirB10 subunits[7,17], whereas those of expanded systems typically have greater subunit complexity. Among the structurally characterized OMCCs, those of minimized systems exhibit uniform 14-fold symmetry, but those of expanded Dot/Icm and Cag systems consist of substructures with different symmetries[2,13–16]. The symmetry mismatches in these latter systems appear to be accommodated through different structural motifs, but how machine asymmetries contribute to T4SS$_F$ function is not defined.

The T4SS$_F$ mediates high-frequency transfer of IncF plasmids and accounts for the prevalence of these multi-drug resistance plasmids in environmental and clinical settings. The T4SS$_F$ also is the only T4SS shown to date to produce pili capable of dynamic extension and retraction[9,18]. T4SS$_F$ nanomachines were recently visualized at a resolution of ~2.3 nm in the native context of the bacterial cell envelope by in situ cryoelectron tomography (CryoET)[19]. These in situ images identified certain structural features of the T4SS$_F$, such as the OMCC$_F$ attached to the OM, a central cylinder spanning the periplasm to the inner membrane (IM), and an inner membrane complex (IMC) dominated at its cytoplasmic interface by the hexameric VirB4-like ATPase TraC. The OMCC$_F$, with an overall width of ~250 Å and a 13-fold-symmetrical outer-lobed ring and an inner ring of undefined symmetry, appears to form a pore across the OM for substrate transfer and also serves as a basal platform for the dynamic F pilus. These features strongly indicate that the OMCC$_F$ plays a central role in regulating transitions between the distinct functional states of the T4SS$_F$.

In this work, we report the structure of the OMCC$_F$ solved at ~3.0 Å resolution by single-particle cryoelectron microscopy

(CryoEM). The near-atomic models reveal that the OMCC$_F$ is composed of only three subunits, VirB7-like TraV, VirB9-like TraK, and VirB10-like TraB, reminiscent of the minimized systems. Remarkably, the OMCC$_F$ is configured as two distinct substructures, an outer ring complex (ORC) with 13-fold symmetry and a central cone (CC) with 17-fold symmetry, and thus exhibits the property of symmetry mismatch shared by the expanded systems. VirB7-like TraV and VirB10-like TraB accommodate the symmetry mismatch by binding both substructures, a property designated here as an intraprotein symmetry alteration. We propose a model in which the stable binding of one domain of TraV and TraB to one substructure and dynamic binding of a second domain to subsets of sites on the mismatched substructure impart conformational flexibility to the OMCC$_F$ required for activation of the T4SS$_F$ for F plasmid transfer or F pilus production.

## Results

**Overall architecture of the T4SS$_F$ OMCC.** We determined an asymmetric structure (C1 reconstruction) of the F-encoded OMCC (OMCC$_F$) at 4.22 Å resolution from 70,700 purified particles (Fig. 1a). The particles were purified from *E. coli* harboring the IncFV plasmid pED208 engineered to carry a Strep-tag sequence at the 3′ end of *traB*. This strain stably produces TraB-Str and elaborates fully functional T4SS machines, as evidenced by wild-type levels of F plasmid transfer and F pilus production (Supplementary Fig. S1a). We used sequential Strep-tag affinity pull-down and size exclusion chromatography to purify complexes (Supplementary Fig. S1b, c). Three dominant species detected in SDS-polyacrylamide gels were identified as TraV,

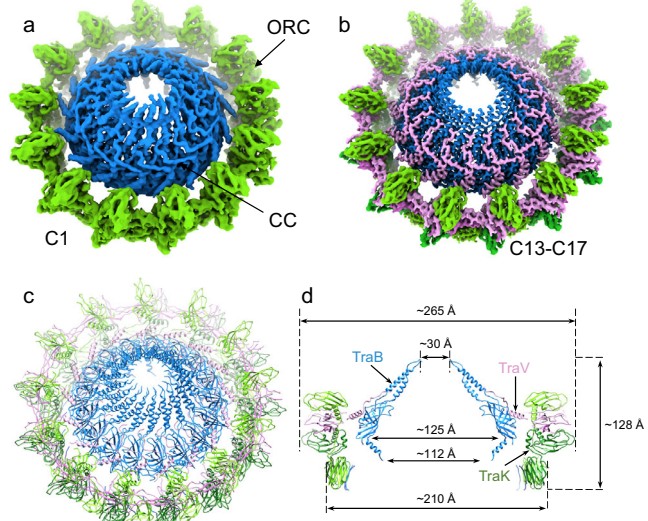

**Fig. 1 Cryo-EM structure of the F-encoded OMCC. a** Asymmetric reconstruction (C1) of the OMCC$_F$ at 4.22 Å (Supplementary Fig. 2). The C1 structure shows two distinct subassemblies, an outer ring complex (ORC) with 13-fold symmetry and a central cone (CC) with 17-fold symmetry. **b** A high-resolution map synthesized from the symmetrized ORC and CC maps. ORC is resolved to 3.31 Å with 13-fold symmetry imposed and the CC is at 2.95 Å resolution with 17-fold symmetry imposed (Supplementary Fig. 2). **c** The atomic models built from (**b**). **d** A central slice of the atomic model (**c**) with different sizes labeled. Subunit colors: TraK (Green), TraB (Blue), TraV (Pink). Structures of the C1, C13, and C17 maps have been deposited with the EMD accession numbers EMD-24768, EMD-24769, and EMD-24770 and atomic models of the C13 and C17 structures have been deposited with the PDB accession numbers 7SPB and 7SPC.

TraK, and TraB by LC-MS/MS mass spectrometry (Supplementary Fig. S1d). Purified material also contained minor amounts of the VirB/VirD4 homologs and several F-specific Tra proteins, but these were not detected in the high-resolution OMCC$_F$ structure. Data were collected and processed as illustrated in Supplementary Fig. S2 flowcharts.

The C1-reconstructed complex is composed of two distinct substructures, the outer ring complex (ORC) with 13-fold symmetry and the central cone (CC) with 17-fold symmetry (Fig. 1a). To enhance the resolution, we performed both C13 and C17 symmetry reconstructions. In the 13-fold symmetry-imposed reconstruction (3.31 Å resolution), the ORC is well-resolved (Supplementary Fig. S3a, c). The CC is blurred due to the application of the wrong symmetry, but the cone shape was readily evident by filtering to a 10 Å resolution. By imposing 17-fold symmetry, we resolved the CC at a resolution of 2.95 Å (Supplementary Fig. S3b, c). A blurred density was evident at the top of the CC, which likely consists of outer-membrane (OM) lipids or detergent as this region of the OMCC is known to embed into the OM[17].

In the C13/C17 map, TraK and TraV are the major constituents of the ORC and TraB dominates the CC (Fig. 1b). Models of these subunits were built into the map, resulting in structural definition of the entire OMCC$_F$ (Fig. 1b, c, Table 1). In side-view, the ORC is divisible into upper and lower tiers with outer diameters of 265 Å and 210 Å, respectively (Fig. 1d). In its overall width and 13-lobed architecture, the upper tier of the ORC matches the in situ OMCC$_F$ structure of the T4SS$_F$ visualized in the bacterial cell envelope by CryoET[19]. The CC, which was not well-resolved in the in situ structure, has a

chamber of 125 Å at its widest point that tapers to a 30 Å pore at the top and to 112 Å at the bottom.

The entirety of the OMCC$_F$ isolated from the pED208-carrying strain (wild-type, WT) is composed of TraV, TraK, and TraB. Importantly, the ORC and CC substructures of all assigned classes of OMCC$_F$ particles uniformly displayed 13- and 17-fold symmetries prior to imposing symmetry. We asked whether a strain engineered to produce TraV, TraK, and TraB in the absence of other Tra components also elaborates the OMCC$_F$. Indeed, we were able to isolate OMCC$_F$ particles from the TraV/K/B-producing strain, establishing that other F-encoded Tra proteins are not essential for its assembly. However, in contrast to the OMCC$_F$ particles obtained from the WT strain, those from the TraV/K/B-producing strain were highly heterogeneous in size and symmetry. ORC substructures were grouped into 4 distinct classes with symmetries ranging from 11- to 14-fold, although the 13-fold symmetry class was the most abundantly populated (Supplementary Fig. S4). Reconstruction of the class with 13-fold symmetry imposed resulted in a structure of the ORC at a higher resolution of 2.97 Å than the 3.31 Å resolution achieved for the ORC from the WT strain (Supplementary Fig. S4). For the 13-fold symmetrical ORC class, we further refined the CC and obtained classes with similar numbers of particles exhibiting symmetries of 16-fold or 17-fold. The C16 and C17 CC structures were solved at resolutions of 3.87 Å and 3.54 Å, respectively (Supplementary Fig. S4). As discussed further below, constituents of the OMCC$_F$'s obtained from the WT and TraV/K/B-producing strains align well. However, the observed heterogeneity in OMCC$_F$ particles isolated from the TraV/K/B-producing strain strongly indicates that one or more other pED208-encoded Tra

**Table 1 Cryo-EM data collection, refinement and validation statistics.**

| | WT C1 (EMDB-24768) | WT C13 (EMDB-24769) (PDB 7SPB) | WT C17 (EMDB-24770) (PDB 7SPC) | TraV/K/B C13 (EMDB-24771) (PDB 7SPI) | TraV/K/B C17 (EMDB-24772) (PDB 7SPJ) | TraV/K/B C16 (EMDB-24773) (PDB 7SPK) |
|---|---|---|---|---|---|---|
| *Data collection and processing* | | | | | | |
| Magnification | 81,000 | 81,000 | 81,000 | 81,000 | 81,000 | 81,000 |
| Voltage (kV) | 300 | 300 | 300 | 300 | 300 | 300 |
| Electron exposure (e−/Å²) | 40 | 40 | 40 | 40 | 40 | 40 |
| Defocus range (μm) | −0.8, −2.8 | −0.8, −2.8 | −0.8, −2.8 | −0.8, −2.8 | −0.8, −2.8 | −0.8, −2.8 |
| Pixel size (Å) | 1.0652 | 1.0652 | 1.0652 | 1.0652 | 1.0652 | 1.0652 |
| Symmetry imposed | C1 | C13 | C17 | C13 | C17 | C16 |
| Initial particle images | 323,000 | 323,000 | 323,000 | 321,000 | 321,000 | 321,000 |
| Final particle images | 70,700 | 70,700 | 70,700 | 148,000 | 26,000 | 22,000 |
| Map resolution (Å) | 4.28 Å | 3.31 Å | 2.95 Å | 2.97 Å | 3.56 Å | 3.90 Å |
| FSC threshold | 0.143 | 0.143 | 0.143 | 0.143 | 0.143 | 0.143 |
| *Refinement and model validation* | | | | | | |
| Initial model used (PDB code) | None | None | None | None | None | None |
| Model resolution (Å) | | 3.7 | 3.2 | 3.3 | 4.0 | 4.0 |
| FSC threshold | | 0.5 | 0.5 | 0.5 | 0.5 | 0.5 |
| Model composition | | | | | | |
|   Non-H atoms | | 57,408 | 29,716 | 57,408 | 29,597 | 27,968 |
|   Protein residues Ligands | | 7488 | 3978 | 7488 | 3961 | 3744 |
| R.m.s. deviations | | | | | | |
|   Bond lengths (Å) | | 0.009 | 0.010 | 0.009 | 0.010 | 0.006 |
|   Bond angles (°) | | 1.535 | 1.304 | 1.529 | 1.406 | 1.055 |
| Validation | | | | | | |
|   MolProbity score | | 2.60 | 2.30 | 2.88 | 2.64 | 1.96 |
|   Clashscore | | 19.49 | 12.51 | 16.92 | 17.51 | 14.57 |
|   Poor rotamers (%) | | 4.19 | 3.26 | 9.57 | 6.56 | 2.48 |
| Ramachandran plot | | | | | | |
|   Favored (%) | | 95.02 | 95.61 | 93.58 | 96.04 | 95.75 |
|   Allowed (%) | | 4.45 | 4.39 | 5.43 | 3.96 | 4.25 |
|   Disallowed (%) | | 0.53 | 0 | 0.71 | 0 | 0 |

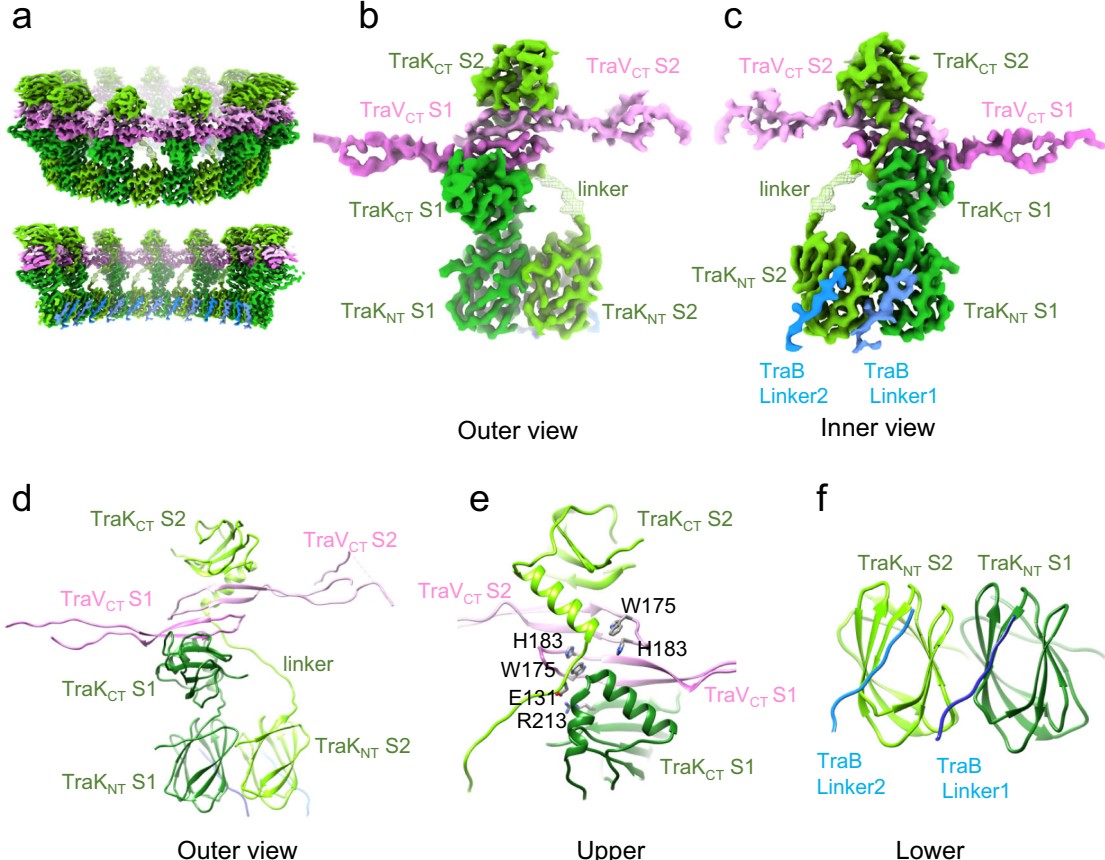

**Fig. 2 Structure of the outer ring complex (ORC). a** Overview (upper) and a central cut view (bottom) of the 3.31 Å ORC from the pED208-carrying (WT) strain reconstructed with 13-fold symmetry imposed. **b, c** Outer and inner views of the heterodimer structural unit including two TraK monomers and two C-terminal domains of TraV (TraV$_{CT}$). Different subunits are denoted S1 and S2. The two N-terminal domains of TraK (TraK$_{NT}$) are arranged side-by-side, and the two C-terminal domains (TraK$_{CT}$) are flipped 180° relative to each other. The two TraV$_{CT}$ domains pack antiparallel to each other. A flexible linker connecting the TraK$_{NT}$-S2 and TraK$_{CT}$-S2 domains is visible as a blurred density at low (10 Å) resolution. Two TraB linker segments corresponding to residues R176-186 form specific contacts with TraK$_{NT}$ domains. **d,e,f** Atomic models of the (**d**) entire structural unit, (**e**) the upper TraK$_{CT}$/TraV$_{CT}$ complex showing residues whose interactions are predicted to stabilize the ORC, and (**f**) the lower TraK$_{NT}$ dimer with associated TraB linker segments. Colors for all panels: TraK-S1 and TraV-S2(dark and light green), TraV-S1 and TraV-S2 (dark and light pink), TraB$_{R176-S186}$ linkers (blue).

subunits play important roles in stabilizing the C13:C17 symmetry of the OMCC$_F$, even though the stabilizing factor(s) was not detected in the OMCC$_F$ map.

**Structure of the outer ring complex.** The ORC is 90 Å in height and 48 Å in thickness, and presents as a distinct upper-lobed ring and a lower-continuous ring (Fig. 2a). It consists of 26 copies of TraK and 26 copies of the C-terminal (CT) domain (residues V150-N204) of TraV (Figs. 1, 2a). Reminiscent of other VirB9 homologs, TraK has two globular domains separated by a flexible linker (Fig. 2b, c). The N-terminal (NT) domain (residues Q25-E120) adopts a β-sandwich fold made of five-stranded and four-stranded β-sheets, and the C-terminal (CT) domain (residues Y136-N238) is comprised of 4 β-strands and one α-helix (α1) that extends away from the β-sandwich (Fig. 2d–f). In the structural unit of the ORC (Fig. S5a), TraK assembles as a dimer with the two TraK$_{NT}$ domains situated side by side at an angle of 13.85° to each other. The two TraK$_{CT}$ domains are flipped 180° relative to each other, and the CT dimer sits laterally on top of the NT dimer (Supplementary Fig. S5a). The TraK$_{CT}$ domains are separated from each other by TraV (see below), although an electrostatic interaction between R213 of TraK subunit 1 (S1) and E131 of subunit 2 (S2) likely participates in stabilization of the TraK$_{CT}$ dimer (Fig. 2e).

The two copies of TraK are therefore configured quite differently in the structural unit. In TraK-S1, the NT and CT domains are juxtaposed, but in TraK-S2 these domains are separated by a distance of ~43 Å (Fig. 2d). These structural differences are afforded by TraK's flexible linker (residues E120-Q134) (Supplementary Fig. S5a). In TraK-S1, the linker is folded and readily detectable in the high-resolution map, but in TraK-S2 it is visible as an elongated density only when the map was filtered to 10 Å resolution (Fig. 2b, c). In the assembled ORC, 13 structural units are arranged in a ring, with the TraK$_{NT}$ domains forming a lower-continuous ring with 26-fold symmetry and the TraK$_{CT}$ domains extending vertically to generate the 13-fold symmetrical, upper-lobed ring (Fig. 2a and Supplementary S4). Although the upper-lobed and lower rings of the ORC exhibit a C13:C26 symmetry mismatch, this symmetry mismatch is readily accounted for by the unique architecture of the TraK dimer in the structural unit. Because only 13 structural units build the entire ORC, we will refer to the ORC as having an overall C13 symmetry, except when specifically discussing the 26 copies of the TraK$_{NT}$ domains.

The C-terminal domain of TraV (TraV$_{CT}$; residues V150-N204) is composed of two antiparallel β-strands connected by a loop (Fig. 2b–e). An interaction between residues W175 and H183 within the intervening loop is expected to stabilize this subdomain. In the structural unit, one TraV$_{CT}$ domain is situated

above and antiparallel to a second. The two TraV$_{CT}$ domains are sandwiched between the two TraK$_{CT}$ domains, configured so that the intervening loops of each TraV$_{CT}$ domain bend inward to form a groove. This groove is occupied by an α-helix situated N-proximally to the CT domain of TraK-S2 (Fig. 2e). This TraK–TraV interaction surface, plus the TraK$_{CT}$-S1 R213–S2 E131 interaction mentioned above, is predicted to stabilize the 3D architecture of the TraK–TraV$_{CT}$ structural unit. In the assembled ORC, the upper and lower TraV$_{CT}$ domains of one structural unit interact with equivalently-positioned TraV$_{CT}$ domains in the flanking structural units, resulting in a two-stranded 'ORC belt' that fixes the TraK$_{CT}$ lobes laterally within the global structure (Fig. 2 and Supplementary Fig. S5b).

In the lower ring of the ORC, we identified additional densities associated with the inner surfaces of each of the 26 TraK$_{NT}$ domains (Fig. 2c, f). By modeling, we confirmed that these densities correspond to residues R176-S186 of TraB, which is part of a long linker (residues K194-Q143) projecting from the C-terminal β-barrel domain through the base of the OMCC$_F$ to the inner membrane. As discussed further below, this finding is important in view of further evidence that 17 copies of the TraB$_{β-barrel}$ domain assemble as the CC. This TraB linker therefore bridges the mismatch between the 26-fold symmetrical lower ring of the ORC and 17-fold symmetrical CC.

The features described above for the ORC substructure of the OMCC$_F$ obtained from the pED208-carrying strain are also evident for the class of C13-symmetrized ORC substructures from the TraV/K/B-producing strain. Structural units from the two strain sources exhibit the same overall architecture, and the TraK and TraV$_{CT}$ constituents also superimpose very well (Supplementary Fig. S6a). The C13-symmetrized ORCs from the WT and TraV/K/B-producing strains can therefore be considered equivalent, although other classes of ORCs with distinct symmetries also assemble in this strain (compare Supplementary Figs. S3, S4).

We tested the functional importance of intersubunit contacts predicted to stabilize the ORC or accommodate the ORC-CC symmetry mismatch (Supplementary Fig. S7). First, we deleted *traV*, *traK*, and *traB* from pED208 and confirmed with complementation tests that each mutation is nonpolar on downstream gene expression. As expected from previous studies of other T4SSs, the Δ*traV* and Δ*traB* mutant strains did not produce functional T4SS$_F$'s, as evidenced by a lack of detectable F pilus production or F plasmid transfer. Although the Δ*traK* mutant also failed to produce F pili, strikingly we found that this mutant retained donor proficiency at a level ($10^{-6}$ Tc'/D) that is well above the threshold of detection (<$10^{-8}$ Tc's/D) (Supplementary Fig. S7a,b). We generated additional mutations in *traK* and *traV* to test the functional importance of predicted interfacial contacts or stabilizing domains, including (i) an Asp substitution for TraK.R213, (ii) Ala substitutions for TraV.W175 and TraV.H183, and (iii) deletion of the entire TraV$_{CT}$ domain (Δ146–204). All of these mutations phenocopied the Δ*traK* mutations in suppressing plasmid transfer by >5–6 orders of magnitude (Supplementary Fig. S7b). These mutations also disrupted F pilus production, as evidenced by an absence of detectable TraA pilin in the extracellular fraction, resistance of the host strain to infection by bacteriophage M13, or lack of F-pilus-mediated aggregation (Supplementary Fig. S7a). Together, these findings support a conclusion that the ORC is essential for production of F pili, but functions mainly to enhance stability or activity of the translocation channel. Remarkably, the CC is capable of mediating DNA transfer across the OM in the complete absence of the ORC.

Next, we deleted the TraB$_{R176-S186}$ linker residues shown to interact with TraK$_{NT}$ domains comprising the lower ring of the

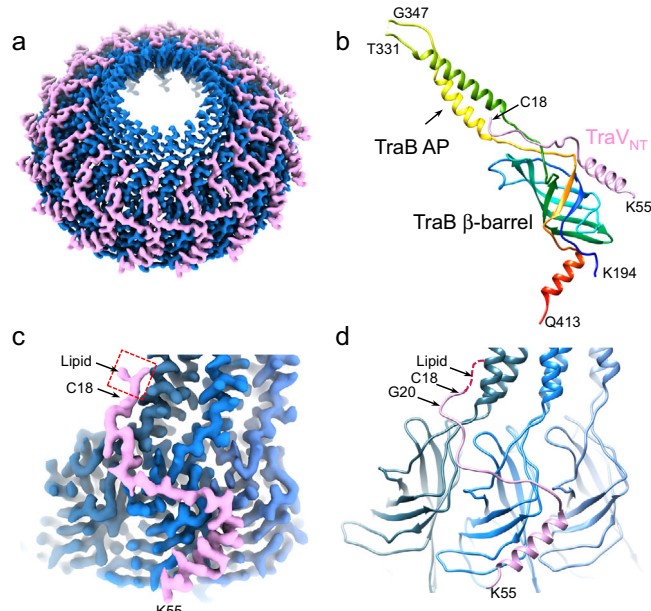

**Fig. 3 Structure of central cone (CC). a** Overview of the 2.95 Å CC from the pED208-carrying (WT) strain reconstructed with 17-fold symmetry imposed. The CC is composed of domains of TraB (blue) and TraV (pink). **b** The atomic model of the β-barrel and AP domains of TraB (residues K194-Q413) in rainbow color, with the N-terminal domain of TraV (TraV$_{NT}$, residues C18-K55) (pink). An AP flexible loop (residues T332-I346) cannot be built in the model. **c, d** Interaction between the TraV$_{NT}$ domain and three adjacent TraB$_{β-barrel}$ domains visualized in densities and models, respectively. The red box in (**c**) shows a piece of lipid density connected to TraV C18.

ORC. A Δ179–185 mutation reduced DNA transfer by ~5 orders of magnitude, and host cells were phage resistant and nonaggregative, indicative of a disruption in F pilus production or stability (Supplementary Fig. S7c, d). By contrast, deletion of residues 186–193, which connect the TraB$_{β-barrel}$ and TraB$_{R176-S186}$ motifs, had no discernible effects on DNA transfer efficiencies or F pilus production. These finding confirm the functional importance of the TraB$_{R176-S186}$–TraK$_{NT}$ interaction in accommodating the ORC/CC symmetry mismatch. Furthermore, the flexible linker (residues 187–193) that was not detected in the CryoEM map, but connects the two TraB domains involved in mismatch accommodation, does not contribute to OMCC$_F$ function.

**Structure of the central cone**. The CC was resolved at 2.95 Å resolution after imposing a 17-fold symmetry on particles purified from pED208-carrying cells (Supplementary Fig. S3b, c). The CC has an outer diameter of ~168 Å and an inner chamber of ~125 Å at its widest point (Fig. 1d). It is composed primarily of 17 β-barrel domains of TraB, a structural motif that is highly conserved among all VirB10 homologs or orthologs (Fig. 3b). An α-helical subdomain termed the antennae projection (AP), which also is highly conserved, extends from the top of each β-barrel. The 17 APs taper in to form a 30 Å pore that is presumed to span the OM. The terminal flexible loops connecting the AP helices could not be completely modeled. At the bottom of the β-barrel, a C-terminal α-helix (residues L398-Q413) projects inward and forms a slight constriction at the base of the central cone.

N-terminal domains of TraV (residues C18-A53) also comprise part of the CC by binding laterally along the surfaces of three TraB$_{β-barrel}$ domains (Fig. 3). In the assembled cone, adjacent TraV$_{NT}$ domains bind overlapping sets of three TraB β-barrels, forming a lattice predicted to stabilize the CC. Residues C18-N42

bind the TraB β-barrels, and adjacent residues Q43-K55 form an α-helix that protrudes away from the $TraB_{\beta-barrel}$ surface toward the ORC. The N-terminal Cys18 residue is predicted to be lipid-modified, as shown for other VirB7 homologs[20,21]. Correspondingly, we detected an extra density associated with Cys18 residues that probably corresponds to lipid (Fig. 3c, d). This lipid modification is optimally positioned to promote insertion and stabilization of the AP across the OM.

For $OMCC_F$ particles purified from the TraV/K/B-producing strain, the CC substructures associated with ORC classes displaying 11-, 12-, and 14-fold symmetries could not be resolved. However, CCs associated with the C13-symmetrized particles were well-resolved and exhibited either 16- or 17-fold symmetries (Supplementary Fig. S4b). Interestingly, the tertiary structures of $TraV_{NT}$ and $TraB_{\beta-barrel}$ domains comprising the C16- and C17-symmetrical CCs align well with each other, as well as with the equivalent structures comprising the C17-symmetrical CC from the pED208-carrying strain (Supplementary Fig. S6b). These findings reinforce the notion that the CC is conformationally flexible as a result of sparse interfacial contacts connecting the $TraB_{\beta-barrel}$ domains, even to the extent that CCs with different numbers of $TraB_{\beta-barrel}$ domains can assemble in vivo.

We tested the functional importance of the $TraV_{NT}$–$TraB_{\beta-barrel}$ interaction by generating a series of deletions of the $TraV_{NT}$ domain without disrupting the N-terminal Cys18 residue for OM anchoring (Supplementary Fig. S7e, f). Deletion of the entire $TraV_{NT}$ domain (Δ20–55), as well as smaller N-terminal (Δ20–50, Δ20–45, Δ20–40) or C-terminal (Δ35–55, Δ40–55) deletions strongly impaired $T4SS_F$ function. However, removal of the peripherally-docked α-helix (Δ48–55) had no discernible effect. Several contacts distributed along the length of the $TraV_{NT}$ domain exclusive of the α-helix thus appear to be necessary for binding the CC. We had envisioned that the large linker (residues 55–150) connecting the $TraV_{CT}$ and $TraV_{NT}$ domains would impart conformational flexibility between the ORC and CC of functional importance. Remarkably, however, a Δ54–155 mutation had no effect on either DNA transfer or F pilus production (Supplementary Fig. S7e, f).

**TraB and TraV accommodate the ORC/CC symmetry mismatch.** Reconstruction of the asymmetrical structural unit of the $OMCC_F$ supplied further insights into how TraB and TraV accommodate the C13:C17 mismatch between the ORC and CC substructures. In the case of TraB, the β-barrel domain is the major element of the CC and the R176-S186 linker segment binds $TraK_{NT}$ domains constituting the lower ring of the ORC (Fig. 4a, b). We identified several important features of the $TraB_{R176-S186}$–$TraK_{NT}$ interaction (Fig. 4b and Supplementary S8). By examination of the C13 structure, all sites on the 26 $TraK_{NT}$ domains appear to be occupied by $TraB_{R176-S186}$. However, the C13 structure represents an averaged structure derived from analyzing thousands of $OMCC_F$ particles, suggesting that the 17 $TraB_{R176-S186}$ segments linked to 17 $TraB_{\beta-barrel}$ components of the CC dock randomly or dynamically with the ORC. In fact, on closer examination of the original densities in the C13 map of $TraB_{R176-S186}$ segments bound to each of the two $TraK_{NT}$ domains in a structural unit, we found that the $TraB_{R176-S186}$ densities invariably are stronger when associated with $TraK_{NT}$ S2 than with $TraK_{NT}$ S1 (Supplementary Fig. S8a). This suggests that $TraB_{R176-S186}$ segments bind one of the $TraK_{NT}$ monomers preferentially or with higher affinity over the second in the assembled ORC. Indeed, the periodicity of $TraB_{R176-S186}$ binding was clearly evident when we superimposed the asymmetrically reconstructed C1 map (Fig. 1a) onto the C13-symmetrized map (Fig. 3a). Only 17 of the 26 sites available on $TraK_{NT}$ domains are occupied by strong $TraB_{R176-S186}$ densities (Fig. 4b, gray densities,

blue linkers). Of the remaining 9 available sites, 7 are unoccupied (right image, magenta linkers) and 2 have very weak densities (magenta linkers denoted with asterisks). This generates an alternating pattern of strong and weaker or nonexistent $TraB_{R176-S186}$ densities among the 26 $TraK_{NT}$ domains (Fig. 4b right, e.g., sites 1, 3, 5, 7, 9, 11, 13,…). In the assembled $OMCC_F$, therefore, 13 of the 17 $TraB_{R176-S186}$ segments stably occupy sites on the 13 TraK-S2 subunits, while the remaining 4 copies of the TraB segments bind subsets of the sites available on the 13 TraK-S1 subunits.

In the asymmetrical structural unit, two $TraV_{CT}$ domains embed into the $TraK_{CT}$ dimer, and long flexible linkers (residues K55-V150) connect to $TraV_{NT}$ domains bound along the surfaces of $TraB_{\beta-barrel}$ domains in the CC (Fig. 4a). In the assembled $OMCC_F$, 26 copies of $TraV_{CT}$ domains comprise the ORC belt, but at most 17 associated NT domains occupy sites on the $TraB_{\beta-barrel}$ domains. We further examined how TraV accommodates the ORC/CC symmetry mismatch by attempting to trace the flexible linkers joining the N- and C-terminal domains. Although the majority of this linker is not traceable, residues R90-G101 were found to associate with the $TraK_{CT}$-S2 domain (Fig. 4a, c). These densities are oriented so that residue R90 is proximal to residue K55, which comprises the C-terminus of the $TraV_{NT}$ domain associated with the $TraB_{\beta-barrel}$ domains. We visualized only 13 $TraV_{R90-G101}$ densities, and each of these extends from the 13 $TraV_{CT}$-S2 domains comprising the upper strand of the ORC belt. These findings support a proposal that the 13 $TraV_{NT}$-S2 domains connected to the traceable R90-G101 densities occupy 13 of the 17 available sites in the CC.

To reconcile this model with the observation that all 17 sites appear to be occupied in the C17-symmetrized map (Fig. 1), we postulated that the 13 $TraV_{NT}$-S2 domains bind dynamically to 13 of the available 17 sites on the CC. Indeed, when we compared density intensities of the $TraV_{NT}$ domains to those of $TraB_{\beta-barrel}$ domains in the C17 map, we found that the TraV densities are weaker than the TraB densities, suggestive of subsaturated occupancy of available sites. There also are no variations in the $TraV_{NT}$ density intensities among the 17 sites on the CC in the C1 reconstruction (Fig. 1a), further indicating that, unlike the $TraB_{R176-S186}$ segments, the $TraV_{NT}$-S2 domains bind with comparable affinities to all available sites on $TraB_{\beta-barrel}$ domains (Fig. 4c right, blue NT domains). We were unable to detect the 13 $TraV_{NT}$ domains and linkers associated with $TraV_{CT}$-S1 domains comprising the lower strand of the belt, and therefore it remains possible that 4 $TraV_{NT}$-S1 domains occupy the remaining sites on the CC. The $TraV_{NT}$-S1 domains additionally or alternatively might assemble at the periphery of the ORC to anchor this substructure to the OM through N-terminal lipidation or perhaps recruit other Tra proteins of functional importance to the $OMCC_F$. In line with this latter possibility, when the cryoEM structure of the $OMCC_F$ was superimposed onto the structure of the $T4SS_F$ generated previously by in situ cryoET[19], the periphery of the ORC closely associates with the OM, possibly through connections mediated by the acylated N-termini of the $TraV_{NT}$-S1 domains (Supplementary Fig. S9a). These peripheral connections, and those contributed by the lipid-modified $TraVNT$-S2 domains associated with the CC, anchor the $OMCC_F$ at the periplasmic face of the OM. This optimally positions the AP pore, which is the dominant hydrophobic motif of the $OMCC_F$, for insertion into the OM (Supplementary Fig. S9a).

## Discussion
The F plasmid-encoded T4SS is a dynamic bacterial nanomachine that transitions between a quiescent complex and activated states mediating F pilus assembly or plasmid transmission[2,19,22]. The near-atomic resolution structures presented here advance an

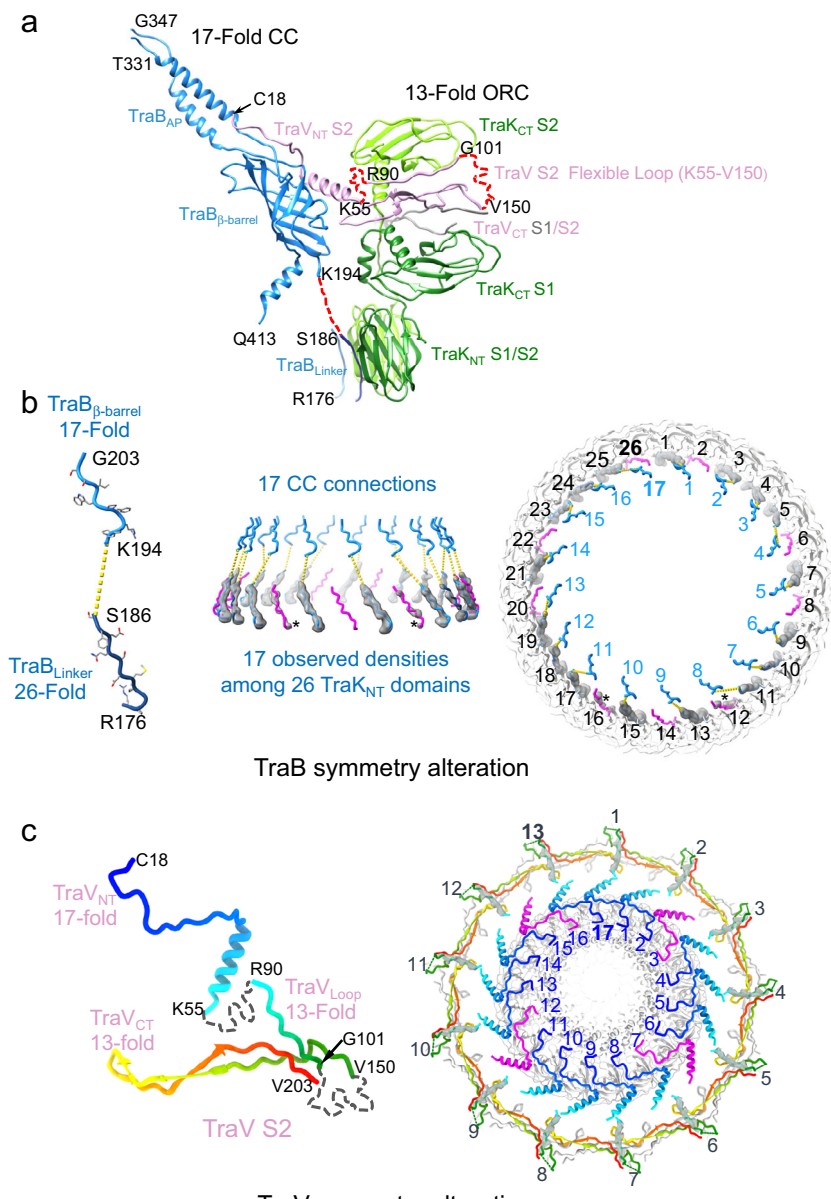

**Fig. 4 TraV and TraB accommodate the C13:C17 symmetry mismatch. a** An asymmetric unit of the OMCC$_F$. Thirteen TraK/TraV$_{CT}$ units comprise the C13 ORC, while the 17 TraB$_{\beta\text{-barrel}}$/TraV$_{NT}$ units assemble as the C17 CC. TraV and TraB bridge the ORC and CC substructures through flexible loops (red dash lines). Subunit domains and colors are shown as described in Fig. 2 legend. Residues shown denote boundaries of the different domains. A short segment (residues R90-G101) in the TraV S2 flexible linker forms a traceable contact with the TraK$_{CT}$ S2 domain. Two TraB$_{R176\text{-}S186}$ linker segments are shown to denote detectable contacts with both S1 and S2 domains of the TraK$_{NT}$ dimer. **b** Left: Segment of TraB involved in the intraprotein symmetry alteration. Residues K194-G203 are part of the TraB$_{\beta\text{-barrel}}$ domain, 17 copies of which form the CC. Residues S186-K194 bridge the symmetry mismatch, but can be deleted without affecting T4SS$_F$ function. Residues R176-S186 bind a subset of the 26 copies of TraK$_{NT}$ domains comprising the ORC. Middle and Right: Schematic views depicting the TraB symmetry alteration from side and top, respectively. Color-coding is as shown at left. Densities (gray) are extracted from the C1 reconstruction, which shows 17 strong densities (blue sites), 7 empty densities (magenta sites), and 2 weak densities (* at sites #12 and 16, magenta). Numbers refer to the 26 TraK$_{NT}$ domains and 17 connections to the TraB$_{\beta\text{-barrel}}$ domains. **c** Left: TraV monomer (Rainbow color) displays an intraprotein symmetry alteration. A subset of the 26 TraV$_{NT}$ (residues C18-K55) bind 17 available sites in the CC. The flexible loop (residues K55-V150) bridges the symmetry mismatch, but can be deleted without affecting T4SS$_F$ function. Twenty-six copies of the TraV$_{CT}$ domain (residues V150-N204) comprise the ORC belt. Residues R90-G101 in the flexible loop associated with TraV S2 bind the 13 copies of TraK$_{CT}$-S2 domains. Right: Schematic view depicting the TraV intraprotein symmetry alteration viewed from top. Densities associated with the traceable R90-G101 segment from TraV S2 are shown. The 4 magenta-colored TraV$_{NT}$ domains (randomly selected) have no traceable paths to TraV$_{CT}$ domains. The 13 TraV$_{NT}$ S2 domains (blue) are connected via the flexible loop to the TraV$_{CT}$-S2 domains (rainbow). TraB$_{\beta\text{-barrels}}$ comprise the CC (gray, inner ring). TraV$_{NT}$-S1 domains connected to the TraV$_{CT}$-S1 domains (gray, outer ring) have not been identified.

understanding of how the $OMCC_F$ orchestrates the structural transitions associated with these different functional states. Remarkably, the $OMMC_F$ is composed of ORC and CC substructures exhibiting C13 and C17 symmetries, respectively, and TraV and TraB accommodate the symmetry mismatch by binding both elements through a mechanism designated as an intraprotein symmetry alteration. The $TraV_{NT}$ and $TraB_{\beta\text{-barrel}}$ domains stably comprise major parts of the ORC and CC substructures, but our data suggest that the $TraV_{CT}$ and $TraB_{R176\text{-}S186}$ domains bind variably or dynamically to the mismatched substructures. These results support the following dynamic model for the action of the $OMCC_F$. The $T4SS_F$ is initially assembled in a quiescent state, but is activated by unknown signals to elaborate F pili or by signals such as plasmid docking at the channel entrance and target cell binding for substrate transfer[23,24]. Accompanying machine activation, $TraV_{NT}$ and $TraB_{R176\text{-}S186}$ domains are triggered to dynamically bind subsets of available target sites in the CC and ORC. Such reiterative binding to adjacent sites might promote lateral or ratcheting movement of the CC relative to the ORC to power cycles of TraA pilin insertion and deinsertion during F pilus extension and retraction. Alternatively, dynamic binding might induce vertical insertion and deinsertion of the CC into and from the OM for reiterative rounds of substrate passage. Further analyses of $T4SS_F$'s in different states are clearly needed to test these ideas and define the nature of structural transitions associated with machine activation. By solving the $OMCC_F$ structure likely in its quiescent state, and by defining the importance of key structural elements for the myriad of $T4SS_F$ functions, our present findings have laid the groundwork for such future investigations.

The $OMCC_F$ particles purified from the pED208-carrying strain displayed a uniform C13:C17 architecture and most $OMCC_F$ particles recovered from the TraV/K/B-producing strain also displayed this symmetry mismatch. From the TraV/K/B-producing strain, however, $OMCC_F$ complexes with different ORC/CC symmetries were also abundantly represented. When produced in the absence of other pED208-encoded factors, TraV, TraK, and TraB can thus pack in variations of the 26:26:17 stoichiometries observed for WT $OMCC_F$ complexes, supporting our proposal that other pED208-encoded factors stabilize the C13:C17 architecture of the WT complexes. Other VirB-like subunits, such as those comprising the IMC or central channel, might act as stabilizing elements. It is also noteworthy that F plasmids encode a variable number of $T4SS_F$-specific subunits of which TraF, TraW, TraH, TraU, TrbB, and TrbC are universally conserved. Prior studies have shown that these factors are required for WT function, localize in the periplasm, and form various interaction networks with each other and TraV, TraK or TraB, together supporting proposals that an F-specific complex(es) binds and regulates the activity of the $OMCC_F$[22,25–27]. While we did not detect F-specific subunits in the solved $OMCC_F$ structure, a recent in situ cryoET structural analyses of $T4SS_F$s assembled within the bacterial envelope revealed the presence of densities possibly contributed by F-specific factors at the junction of the $OMCC_F$ with the OM (see below). Such factors might associate transiently with the $OMCC_F$ to activate the machine in response to signal sensing.

Our mutational analyses confirmed that prominent structural features, including the TraV NT and CT domains and the TraB β-barrel and R176-S186 domains, are essential for plasmid transfer and F pilus production. Our further finding that the flexible intervening linkers $TraV_{54–155}$ and $TraB_{187–193}$ connecting the ORC to the CC are completely functionally dispensable establishes that the two domains of both proteins whose densities clearly contribute to both of the symmetry mismatched substructures can be directly connected. This argues against the possible involvement of truncated proteins or distinct subunit copy numbers in accommodating the symmetry mismatch. Additionally, the flexible linkers do not contribute to the conformational flexibility in the $OMCC_F$ of proposed importance for machine activation. Our dynamic occupancy model postulates instead that the requisite structural transitions are induced by reiterative binding of the $TraV_{NT}$ and $TraB_{R176\text{-}S186}$ domains to sites on the symmetry mismatched substructures. The functional essentiality of the $TraV_{NT}$ and $TraB_{R176\text{-}S186}$ domains and observations that densities of these domains are weaker than those of their $TraB_{\beta\text{-barrel}}$ and $TraK_{NT}$ binding targets in the symmetrized maps together support this model (Fig. 4 and Supplementary Fig. S8).

The notion that the $T4SS_F$ is activated by movement of the CC relative to the ORC needs to be reconciled, however, with the striking discovery that the CC can support a low level of conjugative DNA transfer in mutants lacking TraK or the $TraV_{CT}$ domain and therefore the entire ORC (Supplementary Fig. S7). The CC might retain a residual capacity to undergo the necessary conformational transitions, e.g., insert into and from the OM, even in the absence of the ORC to mediate some intercellular substrate transfer. Interestingly, the ORC-minus mutant machines fail to elaborate F pili, which raises the possibility that the proposed movements of the CC relative to the ORC, driven by dynamic binding of TraV and TraB to mismatched substructures, exclusively drive F pilus biogenesis and not substrate transfer. ΔtraK and other ORC mutations can be grouped among a class of previously-described uncoupling mutations that selectively disrupt pilus production without blocking assembly of functional translocation channels[28–30]. Such Tra⁺, Pil⁻ uncoupling mutations have been isolated in various T4SS subunits, including the TraK-like VirB9 component of the A. tumefaciens VirB/VirD4 T4SS[29]. These genetic findings establish a precedent for the selective contributions of VirB9 subunits, which invariably are configured as peripheral scaffolds of OMCCs[2], for pilus production.

The WT $OMCC_F$ structure provides unprecedented detail of a $T4SS_F$ subassembly in a closed or quiescent state, as deduced by the absence of an associated F pilus or target cell contact. Deciphering the nature of structural changes accompanying machine activation remains a formidable challenge using single-particle CryoEM approaches. However, the recent visualization of $T4SS_F$ nanomachines in their native cellular context by in situ CryoET has enabled comparisons of quiescent and pilus-bound structures[19]. These studies have generated important insights into the nature of conformational changes, particularly at the $OMCC_F$–OM junction, associated with pilus biogenesis. In the quiescent channel, the $OMCC_F$–OM junction is characterized by invagination of the OM and the absence of discernible OM-spanning densities, consistent with a closed state. By contrast, in the F pilus-bound structure, distinct disc and plug densities are evident at the distal end of the $OMCC_F$ and within the OM. Such densities might be contributed the F-specific Tra components shown to be required for machine activation. Additionally, in the quiescent channel, a ~6 nm central cylinder extends from the IM to the base of the $OMCC_F$, where it then widens to ~9 nm. In the pilus-generating machine, this cylinder does not widen at the base of the $OMCC_F$, but instead projects up through the entire $OMCC_F$ to the OM, where it constricts to ~4 nm. The cylinder, whose composition is presently unknown[19], is therefore a dynamic structure that imparts conformational changes in the $OMCC_F$ required for F pilus production. These findings strongly indicate that the $OMCC_F$ is subject to active remodeling when the $T4SS_F$ initiates pilus production.

At this time, OMCCs from five functionally-distinct T4SSs have been solved at or near atomic resolution, allowing for

detailed comparisons (Supplementary Fig. S9b)[11–16]. Although the $OMCC_F$ resembles the equivalent complexes from the pKM101 Tra and *X. citri* VirB/VirD4 minimized systems in its three-subunit composition, its symmetry mismatch and requirements for other F-specific Tra subunits for machine function support classification of the $T4SS_F$ as a member of the expanded systems. The $OMCC_F$ is also distinct from those of the *H. pylori* Cag and *L. pneumophila* Dot/Icm T4SSs, however, in its smaller size, reduced compositional complexity, and types of symmetry mismatches (Supplementary Fig. S9b). With respect to the latter property, we identified two symmetry mismatches in the $OMCC_F$. The first is the C13:C26 mismatch between the upper-lobed and lower ring of the ORC, which is readily accounted for by the unique architecture of the TraK dimer. In the Cag and Dot/Icm systems, C- and N-terminal domains of the VirB9 homologs (CagX and DotH) also comprise parts of upper and lower sub-structures, respectively termed the outer-membrane cap (OMC) and periplasmic ring (PR). However, the OMC and PR sub-structures are compositionally more complex than the ORC, and they also display symmetry mismatches that are distinct from the C13:C26 symmetries exhibited by the upper and lower ORC rings (Supplementary Fig. S9b). A recent study also has shown that VirB9-like DotH accommodates the OMC:PR symmetry mismatch in the Dot/Icm system not through unique configurations of DotH dimers or by intraprotein symmetry alterations, but rather through interactions of distinct numbers of DotH subunits with the two mismatched substructures[16].

The second symmetry mismatch in the $OMCC_F$ is between the C13 ORC and C17 CC. The corresponding region of the Cag system is uniformly 14-fold symmetric and composed of 14 copies each of VirB9-like $CagX_{NT}$ and VirB10-like $CagY_{\beta-barrel}$ domains, plus a number of other system-specific subunits (Supplementary Fig. S9b)[13,15]. In the Dot/Icm system, the outer portion of the OMC and a central region designated as the central dome are composed of VirB9-like $DotH_{NT}$ and VirB10-like $DotG_{\beta-barrel}$ domains, respectively, but the OMC/dome exhibits a C13:C16 symmetry mismatch[14,16]. As shown here for TraV, the DotC lipoprotein also bridges both complexes, suggesting that a TraV-like intraprotein symmetry alteration might accommodate this mismatch. Finally, it is interesting to note that the $TraB_{R176-S186}$–$TraK_{NT}$ interaction detected in the $OMCC_F$ complex is also conserved in the Cag and Dot/Icm systems (Supplementary Fig. S8b). In both systems, N-terminal domains of the VirB9-like subunits comprising the PR substructures assemble as dimers, and linker segments of the VirB10-like subunits extending from the β-barrel domains form specific contacts with these $VirB9_{NT}$ dimers. It is enticing to suggest that the dynamic occupancy model proposed here for the $TraB_{R176-S186}$ linker segment also serves to accommodate the OMC/dome:PR asymmetries in the Cag and Dot/Icm OMCCs.

In summary, the $OMCC_F$ appears to represent a hybrid of the equivalent substructures associated with minimized and expanded T4SSs. It is also structurally distinct from other OM-bound substructures of other bacterial nanomachines dedicated to export of macromolecules. Recent studies have identified symmetry mismatches among a growing list of such nanomachines, including members of the type II (T2SS), type III (T3SS), and type VI (T6SS) secretion superfamily, the flagellar motor, and $F_1F_o$-ATP synthase[31–36]. Although symmetry mismatches are thought to impart conformational flexibility, the mechanisms by which mismatches are accommodated and their biological importance have not been fully defined. The mechanisms by which TraV and TraB bridge the ORC and CC substructures thus potentially informs an understanding of how symmetry mismatches are accommodated in other bacterial nanomachines. The $OMCC_F$ is also considerably compositionally and structurally simpler than

these other bacterial nanomachines, making it an ideal complex for future investigations exploring the structural bases and biological consequences of intrinsic symmetry mismatch. From a broader medical perspective, structural definition of the $OMCC_F$ and of other OMCCs associated with T4SSs paves the way for design of small molecule inhibitors of these nanomachines of potential value for therapeutic intervention of multi-drug resistance proliferation and virulence.

## Materials and methods

**Bacterial strains and growth conditions.** Bacterial strains used in this study are listed in Supplementary Table S1. *E. coli* strains were grown at 37 °C with shaking in Lysogeny Broth (LB) unless otherwise indicated. Plasmids were maintained by selection with the following antibiotics with concentrations in parentheses: carbenicillin (100 µg ml$^{-1}$), spectinomycin (100 µg ml$^{-1}$), chloramphenicol (20 µg ml$^{-1}$), tetracycline (20 µg ml$^{-1}$), kanamycin (50 µg ml$^{-1}$).

**Construction of pED208-$traB_{Strep4}$ and deletions of *traV*, *traK*, and *traB*.** The *traB* gene along with flanking ~600 bp upstream and downstream sequences was cloned in pBAD24. This construct was further used to incorporate a double twin-Strep tag ($Strep_4$) epitope at the 3′ end of *traB* by inverse PCR using the oligonucleotides listed in Supplementary Table S1. Next, $traB_{Strep4}$ was amplified and incorporated in pED208Δ*traB*::FRT-Kan$^r$-FRT by λ *red-gam* induced homologous recombination to excise the FRT-Kan$^r$-FRT cassette and yield pED208-$traB_{Strep4}$[37,38]. Incorporation of the 4xStrep tag at the end of *traB* on pED208 was confirmed by sequencing. pED208-$traB_{Strep4}$ produced stable $TraB_{Strep4}$ and was phenotypically identical to wild-type pED208 with respect to its roles in DNA transfer and F pilus production (Supplementary Fig. S1). *E. coli* strain HME45(pED208) was used for generating Δ*traB*, Δ*traK* and Δ*traV* mutants[37–39]. A FRT-Kan$^r$-FRT cassette from pKD13[40] was amplified with primers listed in Supplementary Table S1 to carry flanking ~50 base-pairs (bps) sequences, homologous to the upstream and downstream regions of the gene targeted for deletion, at the 5′ and 3′ end respectively. The λ *red-gam* system in HME45 was induced by growth at 42 °C. Next, the FRT-Kan$^r$-FRT amplicon was introduced by electroporation, and transformants were selected by plating on LB agar plates supplemented with kanamycin (50 µg/ml). Complementing plasmids were then introduced into respective HME45 strains carrying pED208Δ*traB* (or Δ*traK* or Δ*traV*)::FRT-Kan$^r$-FRT to transfer the mutant pED208 plasmids into MC4100 cells carrying pCP20[40] expressing the Flp recombinase. Transconjugants were streaked on LB agar and grown at 42 °C to induce recombinase expression for excision of the FRT-Kan$^r$-FRT cassette. Growth at 42 °C also cured pCP20. Individual colonies were screened for Chl$^s$ and Kan$^s$ to identify strains with pED208Δ*traB*, Δ*traK* or Δ*traV* plasmids and lacking pCP20. The *tra* mutations were further confirmed by PCR amplification across the deletion junction and sequencing of the PCR fragment.

**Other plasmids.** Plasmid pKBV expressing *traK*, *traB*, and *traV* was constructed by PCR amplification of the corresponding gene cluster from pED208-$traB_{Strep4}$ and insertion into pET-15(b) behind the IPTG-inducible *T7* promoter. Plasmids expressing *traV*, *traK*, and *traB* were generated by PCR amplification of the respective genes and insertion of the amplified gene into pBAD24 downstream of the arabinose-inducible pBAD promoter. These plasmids were used as templates to construct mutant alleles by inverse PCR using oligonucleotides listed in Supplementary Table S1. Mutant alleles were verified by sequencing across the entire genes.

**Purification of the $OMCC_F$.** *E. coli* MC4100(pED208-$traB_{Strep4}$) was grown overnight at 37 °C in LB broth supplemented with appropriate antibiotics. The culture was diluted 1:200 in fresh LB broth without antibiotics and grown at 37 °C with shaking to an $OD_{600}$ of 1.0. *E. coli* BL21(DE3, pKBV) was grown overnight at 37 °C in LB broth supplemented with appropriate antibiotics and then sub-cultured (1:200) in fresh LB broth without antibiotics (1:200) at 37 °C until an $OD_{600}$ of 0.6 was reached. IPTG at a final concentration of 0.1 mM was added and the cells grown at 37°C with shaking for 3 hr. MC4100(pED208-$traB_{Strep4}$) and BL21(DE3, pKBV) cells were harvested and resuspended in ice cold buffer A [50 mM Tris-HCl (pH=8), DNase I (1 µg ml$^{-1}$), lysozyme (200 µg ml$^{-1}$) and EDTA-free protease inhibitor (Thermo Fisher Scientific)]. Cells were lysed on ice using Emulsiflex, then 1 mM each (final concentrations) of EDTA and DTT were added. The cell lysate was centrifuged at 25,000 × *g* for 45 min to remove cell debris, and the supernatant was centrifuged at 95,000 × *g* for 1 h to pellet the membrane fraction. This fraction was solubilized in ice cold buffer B [50 mM Tris-HCl (pH = 8), 200 mM NaCl, 1 mM DTT, 1 mM EDTA, 0.5% DDM, 0.75% DM-NPG, 0.1 % digitonin, EDTA-free protease inhibitor] for 2 h at 4 °C. The suspension was clarified by centrifugation at 25,000 × *g* for 30 min. The supernatant was loaded onto a 1 ml StrepTrap HP (GE Healthcare) column equilibrated with buffer C [50 mM Tris-HCl (pH = 8), 200 mM NaCl, 1 mM DTT, 1 mM EDTA, 0.06% DM-NPG, 0.1 % digitonin]. Following the wash with equilibration buffer, the purified F-T4SS core

complex was eluted in the equivalent wash buffer supplemented with 10 mM D-desthiobiotin and EDTA-free protease inhibitor. The peak fractions were further pooled and loaded onto a Superdex 200 10/300 column (GE Healthcare) equilibrated in elution buffer described above, but without any D-desthiobiotin. Peak fractions containing $OMCC_F$ particles were collected and used for further analysis.

**LC/MS/MS analysis.** Purified complexes were denatured by boiling in Laemmli's buffer, electrophoresed a distance of 1 cm into an SDS-(12%) polyacrylamide gel, and stained with Coomassie blue. After thorough destaining, the stained area of each lane was excised, washed extensively and subjected to thiol reduction by Tris (2-carboxyethyl) phosphine hydrochloride (TCEP) and alkylation with iodoacetamide. Samples were digested overnight with trypsin endoproteinase. An aliquot of the tryptic digest (in 2 % acetonitrile/0.1% formic acid in water) was analyzed by LC/MS/MS on an Orbitrap FusionTM TribridTM mass spectrometer (Thermo ScientificTM) interfaced with a Dionex UltiMate 3000 Binary RSLCnano System. The raw data files were processed using Thermo ScientificTM Proteome DiscovererTM software version 1.4, spectra were searched against the NCBI bacterial database using the Mascot search engine v2.3.02(Matrix Science).

**Conjugation assay.** Conjugation assays were carried out by growth of overnight cultures of donor and recipient strains, dilution (1:50) in fresh antibiotic-free LB broth, and further incubation for 3 h at 37 °C with shaking. Arabinose (0.2% final concentration) was added to induce pBAD-expressed *tra* alleles as needed, and cells were grown at 37 °C for another 1.5 h. Equivalent numbers of donor and recipient cells were mixed and the mating mix was incubated at 37 °C for 1.5 h without shaking. Mating mixtures were then serially diluted and plated onto LB agar containing antibiotics selective for transconjugants (Tc's) and donors (D). The frequency of plasmid transfer is reported as Tc's/D. Donors that failed to transfer the plasmid in broth matings were further subjected to 5 h filter matings. Donor and recipient cultures were mixed in a 1:1 ratio on nitrocellulose filter discs, which were then incubated on an LB agar plate for 5 h at 37 °C. The discs were suspended in LB broth, and the liquid cultures were serially diluted and plated on LB agar containing antibiotics selective for Tc's and donors. All matings were performed at least three times in triplicate, and results are reported for a single representative experiment with Tc/D frequencies for each replicate represented as single dots, mean frequencies by bar heights, and standard deviations by error bars.

**Phage sensitivity assay.** Overnight cultures (50 μl) were spread on LB agar plates containing appropriate antibiotics and arabinose (0.2% final concentration) as necessary for induction of pBAD-expressed *tra* alleles. Cultures were dried, 2 μl of bacteriophage M13 ($10^{11}$ pfu/ml) was spotted, and plates were incubated overnight at 37 °C. Plates were examined for M13-induced formation of turbid plaques.

**Detection of extracellular TraA pilin.** F pili released in the extracellular milieu were recovered and detected by immunostaining as follows. Overnight cultures were diluted (1:200) in fresh LB without antibiotics and when cultures reached an $OD_{600} = 0.6$, arabinose (0.2%, final concentration) was added as necessary for induction of pBAD-expressed *tra* alleles. Following growth for another 2 h, cells were vortexed vigorously and centrifuged at $11,200 \times g$ for 3 min at room temperature. The supernatant was centrifuged twice more to eliminate bacterial cells. The supernatant was then mixed 9:1 with 10X PEG buffer (2% PEG8000, 0.55 M NaCl final concentrations), and the mixture was incubated for 30 min at room temperature and then centrifuged at $18,200 \times g$ for 30 min. The supernatant was removed carefully and the pellet was resuspended in PBS. Samples were subjected to SDS-polyacrylamide gel electrophoresis (SDS-PAGE) and immunostaining of proteins transferred to nitrocellulose membranes using primary antibodies against pED208-encoded TraA pilin (kind gift from Laura Frost) or the β subunit of *E. coli* RNA polymerase (RNA pol) as a control for cell lysis. Anti-TraA antibodies are available upon request; anti-RNA pol antibodies are commercially available from Biolegends.

**CryoEM sample preparation and data collection.** Aliquots (4.0 μl) of purified protein complexes were deposited on glow-discharged ultrathin carbon film on Lacey 300 mesh Au grids. The deposited samples were allowed to stay on grids for 30 sec in the FEI Vitrobot chamber maintained at 23 °C with 100% humidity. Grids were blotted for 4 sec using a blot force of −11 and rapidly plunged in liquid ethane for vitrification. The data were collected using an FEI Titan Krios transmission electron microscope, operated at 300 kV and equipped with Gatan GIF Quantum energy filter. The micrographs were acquired with EPU software (FEI), recorded on a K2 Summit direct electron detector (Gatan) operated in counting mode with a physical pixel size of 1.07 Å per pixel. The detector was placed at the end of a GIF Quantum energy filter operated in zero-energy-loss mode with a slit width of 20 eV. The total exposure time was 7 sec and intermediate frames were recorded every 0.2 sec giving an accumulated dose of ~40 e⁻/Å² and a total of 35 frames per image. The defocus range used was from −0.8 μm to −2.8 μm.

**CryoEM data processing.** Cryosparc v.3.1.0 was used to process the data, including motion correction, CTF correction, particle picking, 2D classification, initial model building, 3D classification, and final refinements[41]. In general, we first used a small set of micrographs to pick particles and build a reliable low resolution map, which was used to create 2D projection images as templates for particle picking of the whole dataset (Fig. S2, flowchart 1). After having the templates, we followed the procedure illustrated in Fig. S2, flowchart 2 to process the dataset to final structures.

**Generate 2D template for particle picking.** Initially 7,000 particles were picked from 200 micrographs using cryoSparc's blob particle picking procedure with setting the blob diameter to 500 Å[41]. We performed 2D classification for the 7,000 particles, A subset of particles were selected and used as templates to re-pick particles from the 200 micrographs. After another 2D classification, 1,800 particles were selected for ab initio 3D reconstruction. Three ab-initio C1 symmetry models were built, two models show 13-fold symmetry (1,600 particles). With these particles, we built a 13-fold initial model, and then refined it to an 8 Å map. Based on this map, 25 uniformly distributed 2D projection images were created as templates for subsequently particle picking.

**CryoEM 3D reconstructions.** For the $OMCC_F$ from the pED208-carrying strain, a total of 323,000 particles were extracted from 8,500 micrographs through cryoSparc's template particle picking procedure. We performed 2D classifications and 150,000 particles contributing to the best 2D classes were selected. C1 symmetry ab initio 3D model building was performed to generate 5 classes. After removing classes that showed broken outer rings, 124,000 particles with complete (non-broken) outer ring density were selected. C13 symmetry was then imposed for refinement. Duplicating this better resolved ORC map as 5 initial models, we performed C1 heterogeneous refinement for the 124k data. The best classes of 70,700 particles that exhibited uniform density for both the ORC and CC were selected. With these particles, we further performed C1, C13, C17 refinements. The corresponding maps were resolved at resolutions of 4.28 Å, 3.31 Å, 2.95 Å, respectively. By applying all possible sorting strategies, we determined that the $OMCC_F$ purified from the pED208-carrying strain contains only a single symmetry configuration: 13-fold for the ORC, 17-fold for CC.

For the OMCC from TraV/K/B-producing strain, a total of 321,000 particles were extracted from 5,500 micrographs using the 2D template images created from analyses of OMCC particles from the pED208-carrying strain. An initial 2D classification showed 14-fold and 13-fold symmetries in the top views of the 2D classes. 240,000 particles contributing to the best 2D classes were selected and used for ab initio 3D model building with 5 classes. Four types of symmetries were identified, ranging from 14-to 11-fold. Then symmetrized model refinements were performed for each subdataset. Using these better resolved ORC structures with different symmetry numbers of 14, 13, 12, or 11 as initial models, multiple rounds of heterogeneous refinement were performed to further sort out datasets into different symmetries sub-datasets. The sub-datasets with 14-fold, 13-fold, 12-fold, 11-fold symmetries contain 34k, 148k, 38k, 20k particles, respectively. After symmetrized refinements for each symmetry, resolutions of 5.28 Å, 2.97 Å, 4.46 Å, 6.9 Å were achieved for the 14-, 13-, 12-, and 11-fold complexes, respectively. We performed C1 refinement for the C13 ORC particles. To further resolve the CC of the C13 ORC structure, ab initio C1 refinement was performed with 5 classes. One subdataset (65k particles) with high quality ORC and CC structures was subjected to heterogeneous refinements with 3 classes using the corresponding ab initio C1 model. The CCs of one class exhibited 17-fold symmetry (22k particles), of a second class exhibited 16-fold symmetry (26k particles), and of a third exhibited an ambiguous symmetry (17k particles). The first two classes were further refined with 17- and 16-fold symmetry imposed and the maps were used as two models for heterogeneous refinement. The final C17 (26k particles) and C16 (22k particles) reconstructions have resolutions of 3.56 Å and 3.9 Å respectively.

**Model building and refinement.** Maps were initially sharpened using the Auto-sharpen option in Phenix v.18.2[42]. The sequences of the structural proteins were first submitted to the SWISS-MODEL SERVER[43] (https://swissmodel.expasy.org), from which initial models were produced for the purpose of initiating 3D coordinates of the amino acids of each molecule. These models were then fitted into the density maps using "Fit in Map" option in UCSF ChimeraX v.1.2[44]. Manual refinement of the model to the density was then performed using Coot v.0.9.4.1[45]. At this stage, fit to density was prioritized, matching secondary structure and large, bulky side chains in the density, providing an initial coarse grain model for subsequent optimization. After coarsely fitting the models in Coot, real-space refinement was performed with 'phenix.real_space_refine' and subsequently adjusted again manually in Coot. This process was repeated in order to maximize fit to density, minimize Ramachandran angle outliers and eliminate steric clashes. Once individual protein subunits were refined, an asymmetric unit was then created and optimized with the same iterative refinement procedure. A full complex with NCS symmetry was then created in Chimera, followed by a final round of Phenix real-space refinement. Final model statistics were calculated in Phenix using the "Comprehensive Validation" tool in Phenix and reported in Table 1. Map-model visualization was performed in Coot and UCSF Chimera.

**Reporting summary**. Further information on research design is available in the Nature Research Reporting Summary linked to this article.

## Data availability

All cryo-EM density maps included in this manuscript are available through the Electron Microscopy Data Bank with accession codes EMD-24768 (WT $OMCC_{C1}$), EMD-24769 (WT $OMC_{C13}$), EMD-24770 (WT $CC_{C17}$), EMD-24771 (TraV/K/B $OMC_{C13}$), EMD-24772 (TraV/K/B $CC_{C17}$) and EMD-24773 (TraV/K/B $CC_{C16}$). The atomic coordinates have been deposited in the Protein Data Bank with accession codes 7SPB (WT $OMC_{C13}$), 7SPC (WT $CC_{C17}$), 7SPI (TraV/K/B $OMC_{C13}$), 7SPJ (TraV/K/B $CC_{C17}$) and 7SPK (TraV/K/B $CC_{C16}$). Source Data are included with this paper. Source data are provided with this paper.

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

## Acknowledgements

This work was funded by Welch Foundation award AU-1953-20180324 and NIH 1R35GM138301 to B.H., NIH 1R35GM131892 and R01GM48746 to P.J.C, and NIH R21AI142378 to B.H. and P.J.C. We acknowledge the Texas Advanced Computing Center (TACC) at The University of Texas at Austin for providing computing training resources to X.L. We gratefully thank the UTHealth Houston CryoEM Core for use of their facilities and microscopes. We thank Laura Frost for her generous gift of anti-TraA antibody specific to pED208-encoded TraA. We thank Li Li and the Clinical and Translational Proteomics Service Center for the mass spectrometry analyses. We thank Rachel Bosserman and Amar Al Mamun for plasmid constructions. We thank members of the Christie and Hu labs for helpful discussions.

## Author contributions

P.K. generated most strains and plasmids, expressed and purified the OMCCs from strains carrying the pED208 or pKBV plasmids, phenotypically characterized mutations, and prepared samples for CryoEM. X.L. processed CryoEM data, built atomic models, interpreted CryoEM results, and prepared figures. M.B. built atomic models. B.H., P.J.C., and X.L. conceptualized the project and designed the research. B.H. interpreted results, prepared figures, and helped craft the manuscript. B.H. and P.J.C. obtained funding for the project. P.J.C. wrote the manuscript with contributions from B.H and X.L. All authors commented on the manuscript.

## Competing interests

The authors declare no competing interests.
