## [Peer Review File · Nature Communications]

Reviewers' Comments:

Reviewer #1:

Remarks to the Author:

Summary:

This manuscript presents the first high resolution structure of the Type IV Secretion System (T4SS) encoded by the conjugative F plasmid. The structure was determined using single particle cryo-EM and provides molecular detail of a previously published lower resolution map of this T4SS in the context of bacteria determined using cryo-electron tomography approaches. This is a very solid structure and presented in an easy to follow and well written manuscript. Interestingly this T4SS seems to be a structural "hybrid" between minimized T4SSs composed of three major components and more complex T4SSs that are composed of 6 or more protein components. Although there is strong structural homology in many of the folds adopted by the species specific components mapped into this structure with other T4SSs, the components themselves are different organizations, different symmetry mismatches, and appear to accommodate these symmetry mismatches in different structural ways. This is very beautiful work and I have only a few minor comments.

Comments:

1. There is a recent structure published of the legionella Dot T4SS published that also characterizes the dome (which seems equivalent to what is referred to as the central cone "CC" in this structure). This is a very recently published manuscript (PMID: 34519271), but it would be worth updating this manuscript and the Supplementary Figure 9 to make sure the most up-to-date structural data for T4SSs are presented.
2. It might also be worth considering adopting the published "dome" nomenclature (versus central cone) used in the Dot T4SS structure to help make it easier for the field to compare/contrast structures— however, this is simply a suggestion and definitely not a required change.
3. The authors should show the C1 reconstruction. Does the C1 structure agree with their model for how the symmetry mismatch is accommodated?
4. Did the authors see any different 3D classes that would indicate different conformations between the CC and the ORC? Did they try using CryoDragon or 3DVA in cryoSPARC? This type of analysis could allow them to see evidence of continuous motion in their structures – providing some structural evidence of their "ratcheting" model for function (which I am not contesting – I think this is a very reasonable model and structural hypothesis).

Minor Comments:

1. In the introduction the authors state that "How symmetry mismatches are accommodated in these systems, and why such asymmetries exist, are unknown." While I agree we don't have high resolution details of how the symmetry mismatches are accommodated in any of these systems or why the symmetry mismatches are so important, the structures of both the *H. pylori* Cag T4SS and the Legionella Dot T4SS actually do show how the symmetry

Reviewer #2:

Remarks to the Author:

The manuscript by Liu et al is a detailed structural analysis of a highly interesting biological nanomachine, the F-plasmid T4SS outer membrane complex. Using state-of-the-art single particle cryo-electron microscopy the authors have solved the structure of this 2.1 MDa proteinaceous complex to a resolution of 3-4 Angstrom. I believe the broad readership of Nature Communications will find this study appealing. I have two suggestions for how the general interest of the study might be increased, but do not suggest that any new experiments need to be done to achieve this. In my opinion the data is of excellent quality and the presentation of the figures and data in the paper is of the appropriate standard for publication.

I see two further questions about the biology of the F-plasmid DNA transfer apparatus that this

structure can help to address:

1. Which elements of the F-plasmid outer membrane complex is embedded in the outer membrane?
2. Does the structure of this F-plasmid outer membrane complex serves as a paradigm for equivalent nanomachines from other bacterial species?

1. Page 12: "... the TraVNT domains also might assemble at the periphery of the ORC to anchor this substructure to the OM through N-terminal lipidation or perhaps recruit other Tra proteins of functional importance to the OMCCF". From the orientation of the N-terminal part of the TraVNT domains, which is where the N-terminal acyl groups would be, it is difficult to envisage how the OMCC is embedded in the outer membrane.

Could an extra panel or panels of analysis be added to either Figure 3 or 4 to show the rim-ward hydrophobic surface on the F-plasmid outer membrane complex, to provide a basis to understand which part of the complex is embedded in the plane of the membrane? Such analyses are commonly done for smaller nanomachines in the outer membrane, such as the BAM complex (doi: 10.1038/nature12521) and T2SS (doi: 10.1128/JB.00521-17, doi: 10.1128/mBio.01344-17). It would be of general interest to know how much of the F-plasmid outer membrane complex is exposed in the periplasm (to engage with other subunits of the machinery) and how much is exposed externally. Since the (acylated) cysteine residue that would be proximal to the membrane interface is visible in the cryo-EM maps, it should be possible to determine how much of the OMCC actually sits within the plane of the membrane.

I appreciate that there are assumptions in the literature regarding the positioning in the plane of the membrane, but this structure would be the first chance to test those assumptions.

2. The introduction makes clear that there are simpler versions of the T4SS for which structural information is in hand, and that the structure of the F-plasmid outer membrane complex presented here is relevant to a more complex nanomachine. Could a Figure or Supplementary Figure be added that would depict either (i) the relative distribution of the "simple" and "complicated" T4SS across bacterial species?, and/or (ii) a structure-based comparison of the new F1-OMCC and the OMCCs associated with the 'minimized' (*Xanthomonas citri* VirB/VirD4, pKM101-encoded Tra) and 'expanded' (*Legionella pneumophila* Dot/Icm, *Helicobacter pylori* Cag) systems. Either approach, (i) or (ii), would provide a broad context for the new structure reported here.

Reviewer #3:

Remarks to the Author:

The manuscript "CryoEM Structure of a Type IV Secretion System Core Complex Required for Dissemination of Multi-Drug Resistance F Plasmids" by Liu et al describes the Cryo-EM structure of the outer membrane core complex of the F-type T4SS (specifically the *E. coli* pED208 system).

The authors expressed and purified the OMCC in a native form via a strep tagged traB pED208 plasmid. They confirm the complex was functional y expressed in vivo. They attempted to produce the OMCC TraV/K/B complex in the absence of the other components but found them to be heterogeneous in size and symmetry, though the authors confirm that the Native form and the TraV/K/B forms with the same symmetries are virtually identical.

The authors go on to functionally assess the intersubunit contacts predicted to stabilize the ORC or accommodate the ORC - CC symmetry mismatch. traV, traK, and traB KO mutants were generated and complementation in trans was confirmed. Several site directed mutants in several traV/K/B as well as a deletion of the TraB linker region were generated and tested functionally (F pilus-mediated cellular aggregation, Susceptibility to infection by the F pilus-binding M13 bacteriophage, Immunoblot analyses of extracellular fractions for detection of F pilus subunit TraA, and mating/Plasmid transfer frequencies). It would have been nice to see if any of these mutants produced a function complex (i.e could you purify the complex), to confirm the complex was/was not destabilised, but this is NOT required in the present study. These functional assays really help to interpret the maps and understand how the symmetry mismatches between the inner and outer rings are accommodated.

The authors go on to suggest a model where the ORC and CC symmetry mismatches are bridged by various domains of TarV S2 and TraB. Where appropriate the authors use both their symmetry imposed maps and their C1 map to address the occupancy of the various domains across the different symmetries. The authors present a convincing though not fully resolved model for how the symmetry mismatches are bridged with these two proteins.

This is a well conceived and study which provides valuable insights into an important system. I appreciated effort taken to experimentally/functionally explore some of the the structural findings of this study with appropriate use of controls where required. I think this will be of high interest to the readers of Nature Communication and should be published.

Minor issues:

Line 126 "...which is even higher than..." > "...which is higher than..."

Legend for Fig S7 is unclear: it refers to i, ii, iii but the images are not annotated like this. Consider rephrasing or relabeling for clarity.

Some of the coloring in the structure figures is very similar and hard to tell apart(e.g the shades of green used in TraKS1 vs TraK S2 in figures 2 and 4 [and fig S5 and S6]). Consider using more distinct coloring for different sub-units.

Did the authors attempt any masked refinements or similar – It appears they applied different symmetries but did not apply masks. Masking may improve the outer and inner ring maps (this is a suggestion and would not be required for the manuscript to progress)

RESPONSES TO REVIEWERS' COMMENTS:

Reviewer #1 (Remarks to the Author):

Summary:

This manuscript presents the first high resolution structure of the Type IV Secretion System (T4SS) encoded by the conjugative F plasmid. The structure was determined using single particle cryo-EM and provides molecular detail of a previously published lower resolution map of this T4SS in the context of bacteria determined using cryo-electron tomography approaches. This is a very solid structure and presented in an easy to follow and well written manuscript. Interestingly this T4SS seems to be a structural “hybrid” between minimized T4SSs composed of three major components and more complex T4SSs that are composed of 6 or more protein components. Although there is strong structural homology in many of the folds adopted by the species specific components mapped into this structure with other T4SSs, the components themselves are different organizations, different symmetry mismatches, and appear to accommodate these symmetry mismatches in different structural ways. This is very beautiful work and I have only a few minor comments.

Comments:

1. There is a recent structure published of the legionella Dot T4SS published that also characterizes the dome (which seems equivalent to what is referred to as the central cone “CC” in this structure). This is a very recently published manuscript (PMID: 34519271), but it would be worth updating this manuscript and the Supplementary Figure 9 to make sure the most up-to-date structural data for T4SSs are presented.

RESPONSE: We have updated the Manuscript and Supplementary Figure, with inclusion of the citation for this newly published manuscript in the Reference lists.

2. It might also be worth considering adopting the published “dome” nomenclature (versus central cone) used in the Dot T4SS structure to help make it easier for the field to compare/contrast structures— however, this is simply a suggestion and definitely not a required change.

RESPONSE: We agree it is important to develop a uniform nomenclature for T4SS substructures. However, presently, the central structure comprised of the VirB10 homolog is referred to in various ways (dome, central cone, inner ring, inner layer...) for different T4SSs, and we think the best strategy is to propose a unified nomenclature in a broader reviewer updating the reader on T4SS architectures. We prefer to keep the CC designation, but will definitely adopt the unified nomenclature when it is developed.

3. The authors should show the C1 reconstruction. Does the C1 structure agree with their model for how the symmetry mismatch is accommodated?

RESPONSE: We have shown C1 reconstruction in Fig.1a. Fig 1b is a combination of c13 and c17, which is a synthesized map based on the C1 reconstruction. The densities in Fig4 b and c come from the C1 reconstruction, which shows how the symmetry mismatches are accommodated.

4. Did the authors see any different 3D classes that would indicate different conformations between the CC and the ORC? Did they try using CryoDragon or 3DVA in cryoSparc? This type of analysis could allow them to see evidence of continuous motion in their structures – providing some structural evidence of their “ratcheting” model for function (which I am not contesting – I think this is a very reasonable model and structural hypothesis).

RESPONSE: After extensive 2D and 3D classification, we sorted out 70,700 particles from 323,000 initially picked particles to get the best map. From this small data set, we did not observe different conformations between the CC and the ORC. We agree that will be insightful to sort out continuous conformational changes which will shed light on the working model of the T4SS. To achieve this aim, however, we need to collect massive amounts of data, and then use CryoDragon or cryoSparc 3DVA to sort them out, which is currently being carried out but lies beyond the scope of this paper.

Minor Comments:

1. In the introduction the authors state that “How symmetry mismatches are accommodated in these systems, and why such asymmetries exist, are unknown.” While I agree we don't have high resolution details of how the symmetry mismatches are accommodated in any of these systems or why the symmetry mismatches are so important, the structures of both the H. pylori Cag T4SS and the Legionella Dot T4SS actually do show how the symmetry

RESPONSE: We have revised this statement to indicate that symmetry mismatches in these other systems are accommodated through different structural motifs – this is also expanded on in the Discussion.

Reviewer #2 (Remarks to the Author):

The manuscript by Liu et al is a detailed structural analysis of a highly interesting biological nanomachine, the F-plasmid T4SS outer membrane complex. Using state-of-the-art single particle cryo-electron microscopy the authors have solved the structure of this 2.1 MDa proteinaceous complex to a resolution of 3-4 Angstrom. I believe the broad readership of Nature Communications will find this study appealing. I have two suggestions for how the general interest of the study might be increased, but do not suggest that any new experiments need to be done to achieve this. In my opinion the data is of excellent quality and the presentation of the figures and data in the paper is of the appropriate standard for publication.

I see two further questions about the biology of the F-plasmid DNA transfer apparatus that this structure can help to address:

1. Which elements of the F-plasmid outer membrane complex is embedded in the outer membrane?

RESPONSE: This has not been investigated specifically for the F system, but TraB's β -barrel and antennae projection (AP) are highly conserved among the VirB10 homologs. We and others have analyzed the membrane disposition and biological importance of these structures through mutational analyses and epitope-tagging followed by in situ labeling with antibodies directed against the epitope-tags. The findings support a proposal that AP domains associated with multiple copies of the VirB10 homologs form a pore that spans the OM. The upper surface of the OMCC comprised of the VirB10 β -barrel domains is attached to the inner layer of the OM via the lipidated N-terminal Cys residue of the VirB7 homolog. Now, this is the likely architecture of the machine in the closed, quiescent state, but it is possible that major structural changes occur when the machine is activated for substrate transfer. Nevertheless, by in situ cryoelectron microscopy (cryoET), we have visualized the F-encoded OMCC - OM junction architectures of machines in the quiescent state and when activated for pilus production. Although these images are of lower resolution than the CryoEM structure presented here, they provide valuable insights into which portion of the OMCC embeds into the OM and support the notion that the AP alone spans the OM. To illustrate the proposed architectural arrangement of the OMCC - OM junction, we have now superimposed the new cryoEM structure of the isolated OMCC over the previously visualized in situ cryoET structure - this is presented in a new supplemental figure (Supplementary Fig. S9a).

2. Does the structure of this F-plasmid outer membrane complex serves as a paradigm for equivalent nanomachines from other bacterial species?

RESPONSE: At this time, the OMCCs of 5 T4SSs have been solved at high resolutions ($>3 \text{ \AA}$). All are composed of homologs of the TraK/TraB/TraV subunits, of which the TraB central cone and TraK N- and C-terminal domains are highly conserved structural motifs. Remarkably, however, these 3 subunits adopt different architectures and symmetries among the 5 solved OMCCs. The F OMCC does seem to be an evolutionary intermediate between the 'minimized' and 'expanded' systems and several of its features are common to both types of systems; for these reasons, it might be considered paradigmatic. We highlight the common and distinguishing features among these OMCCs in the Discussion.

1. Page 12: "... the TraVNT domains also might assemble at the periphery of the ORC to anchor this substructure to the OM through N-terminal lipidation or perhaps recruit other Tra proteins of functional importance to the OMCCF". From the orientation of the N-terminal part of the TraVNT domains, which is where the N-terminal acyl groups would be, it is difficult to envisage how the OMCC is embedded in the outer membrane.

RESPONSE: The 13 TraVNT domains that clearly associate with the TraB β -barrel domains in the central cone are perfectly aligned to tether the central cone to the OM, allowing the AP helices to form a pore across the OM. We were unable to visualize the other 13 TraVNT domains, presumably because flexibility afforded by the linker separating the NT from CT domains. The length of this linker certainly allows for these 13 NT domains to dock via the N-terminal acylated cysteine to the OM.

Could an extra panel or panels of analysis be added to either Figure 3 or 4 to show the rim-ward hydrophobic surface on the F-plasmid outer membrane complex, to provide a basis to understand which part of the complex is embedded in the plane of the membrane? Such analyses are commonly done for smaller nanomachines in the outer membrane, such as the BAM complex (doi: 10.1038/nature12521) and T2SS (doi: 10.1128/JB.00521-17, doi: 10.1128/mBio.01344-17). It would be of general interest to know how much of the F-plasmid outer membrane complex is exposed in the periplasm (to engage with other subunits of the machinery) and how much is exposed externally. Since the (acylated) cysteine residue that would be proximal to the membrane interface is visible in the cryo-EM maps, it should be possible to determine how much of the OMCC actually sits within the plane of the membrane.

I appreciate that there are assumptions in the literature regarding the positioning in the plane of the membrane, but this structure would be the first chance to test those assumptions.

RESPONSE: The availability of the in situ CryoET structure of the F-encoded T4SS greatly facilitates interpretation of the in vitro cryoEM structure of the OMCC. The bulk of the OMCC is clearly in the periplasm, and only the distal end, which likely corresponds to the AP helical pore, projects across the OM. As mentioned above, we have added a supplemental figure in which the OMCC cryoEM structure is

superimposed over the in situ T4SS structure; this conveys to the reader how the OMCC is likely configured in relation to the OM in a native cellular context.

2. The introduction makes clear that there are simpler versions of the T4SS for which structural information is in hand, and that the structure of the F-plasmid outer membrane complex presented here is relevant to a more complex nanomachine. Could a Figure or Supplementary Figure be added that would depict either (i) the relative distribution of the “simple” and “complicated” T4SS across bacterial species?, and/or (ii) a structure-based comparison of the new F1-OMCC and the OMCCs associated with the ‘minimized’ (*Xanthomonas citri* VirB/VirD4, pKM101-encoded Tra) and ‘expanded’ (*Legionella pneumophila* Dot/Icm, *Helicobacter pylori* Cag) systems. Either approach, (i) or (ii), would provide a broad context for the new structure reported here.

RESPONSE: Regarding Point (i), there are many examples in which minimized and expanded systems function as conjugation (DNA transfer) machines. Both types of machines have evolved to function in nearly all bacterial species and there doesn’t seem to be a species preference for one or the other type of machines. Regarding Point (ii), we show the different OMCCs solved to date in the last supplemental figure and compare these structures in the discussion. In Supplementary Fig. S9b, we have updated the structure of the OMCC from the *L. pneumophila* Dot/Icm system based on a publication that came out since our original submission; this allows for some additional comparisons between the F and Dot/Icm systems.

Reviewer #3 (Remarks to the Author):

The manuscript “CryoEM Structure of a Type IV Secretion System Core Complex Required for Dissemination of Multi-Drug Resistance F Plasmids” by Liu et al describes the Cryo-EM structure of the outer membrane core complex of the F-type T4SS (specifically the *E. coli* pED208 system).

The authors expressed and purified the OMCC in a native form via a strep tagged traB pED208 plasmid. They confirm the complex was functionally expressed in vivo. They attempted to produce the OMCC TraV/K/B complex in the absence of the other components but found them to be heterogeneous in size and symmetry, though the authors confirm that the Native form and the TraV/K/B forms with the same symmetries are virtually identical.

The authors go on to functionally assess the intersubunit contacts predicted to stabilize the ORC or accommodate the ORC - CC symmetry mismatch. traV, traK, and traB KO mutants were generated and complementation in trans was confirmed. Several site directed mutants in several traV/K/B as well as a deletion of the TraB linker region were generated and tested functionally (F pilus-mediated cellular aggregation, Susceptibility to infection by the F pilus-binding M13 bacteriophage, Immunoblot analyses of extracellular fractions for detection of F pilus subunit TraA, and mating/Plasmid transfer frequencies). It would have been nice to see if any of these mutants produced a function complex (i.e could you purify the complex), to confirm the complex was/was not destabilised, but this is NOT required in the present study. These functional assays really help to interpret the maps and understand how the symmetry mismatches between the inner and outer rings are accommodated.

RESPONSE: We agree, and we are currently working on solving the structures of several mutant OMCCs of interest. We already know that the mutant machines constructed in the present study are detectable by in situ CryoET, so we are confident that OMCCs from these machines are produced and likely sufficiently stable for purification and structural characterization by single particle cryoEM. We will present these structures in our next manuscript.

The authors go on to suggest a model where the ORC and CC symmetry mismatches are bridged by various domains of TraV S2 and TraB. Where appropriate the authors use both their symmetry imposed maps and their C1 map to address the occupancy of the various domains across the different symmetries. The authors present a convincing though not fully resolved model for how the symmetry mismatches are bridged with these two proteins.

This is a well conceived and study which provides valuable insights into an important system. I appreciated effort taken to experimentally/functionally explore some of the the structural findings of this study with appropriate use of controls where required. I think this will be of high interest to the readers of Nature Communication and should be published.

Minor issues:

Line 126 “...which is even higher than...” > “...which is higher than...”

RESPONSE: Change made.

Legend for Fig S7 is unclear: it refers to i, ii, iii but the images are not annotated like this. Consider rephrasing or relabeling for clarity.

RESPONSE: The figure and figure legend now match.

Some of the coloring in the structure figures is very similar and hard to tell apart (e.g. the shades of green used in TraKS1 vs TraK S2 in figures 2 and 4 [and fig S5 and S6]). Consider using more distinct coloring for different subunits.

RESPONSE: We attempted different color options for the two different TraK subunits, and the two shades of green are optimal especially for the ribbon structures. We acknowledge that it is more difficult to distinguish the two TraK subunits in the space-filling structures (Fig. 2b, 2c), but we have extensively labeled these subunits to eliminate any confusion.

Did the authors attempt any masked refinements or similar – It appears they applied different symmetries but did not apply masks. Masking may improve the outer and inner ring maps (this is a suggestion and would not be required for the manuscript to progress)

RESPONSE: To improve the resolution, we used masking skills for both samples from WT and TraV/K/B, for the C1 reconstruction, specifically, radial masks with different gradients were applied. During refinement of ORC and CC, local masks were applied to the ORC or CC.